# Likelihood Ratio Test and the Evidential Approach for 2 × 2 Tables

**DOI:** 10.3390/e26050375

**Published:** 2024-04-28

**Authors:** Peter M. B. Cahusac

**Affiliations:** 1Department of Pharmacology & Biostatistics, College of Medicine, Alfaisal University, Riyadh 11533, Saudi Arabia; pcahusac@alfaisal.edu; 2Department of Comparative Medicine, King Faisal Specialist Hospital and Research Centre, Riyadh 11211, Saudi Arabia

**Keywords:** 2 × 2 table, χ^2^ test, test for independence, likelihood ratio test, G-test, likelihood, odds ratio, data integrity, too good to be true, test for variance, binomial, contingency tables, multi-way tables

## Abstract

Categorical data analysis of 2 × 2 contingency tables is extremely common, not least because they provide risk difference, risk ratio, odds ratio, and log odds statistics in medical research. A $\chi^{2}$ test analysis is most often used, although some researchers use likelihood ratio test (LRT) analysis. Does it matter which test is used? A review of the literature, examination of the theoretical foundations, and analyses of simulations and empirical data are used by this paper to argue that only the LRT should be used when we are interested in testing whether the binomial proportions are equal. This so-called test of independence is by far the most popular, meaning the $\chi^{2}$ test is widely misused. By contrast, the $\chi^{2}$ test should be reserved for where the data appear to match too closely a particular hypothesis (e.g., the null hypothesis), where the variance is of interest, and is less than expected. Low variance can be of interest in various scenarios, particularly in investigations of data integrity. Finally, it is argued that the evidential approach provides a consistent and coherent method that avoids the difficulties posed by significance testing. The approach facilitates the calculation of appropriate log likelihood ratios to suit our research aims, whether this is to test the proportions or to test the variance. The conclusions from this paper apply to larger contingency tables, including multi-way tables.

## 1. Introduction

CHUCK:Well then there’s nothing more to talk about! I will beat this. Ergo, a falsis principiis proficisci. Meaning? (gestures to Saul)

SAUL:That’s the one about false principles, but it’s not…

CHUCK:You proceed from false principles. Your argument is built on quicksand; therefore it collapses.

*Better Call Saul* TV Series (2015–2022)

The 2 × 2 contingency table is commonly encountered in research. For example, it is used in randomized controlled trials where patients are given either treatment or placebo, and the outcome is either healthy or diseased. The design gives rise to risk ratios (relative risk) and odds ratios, which are particularly useful in medical research [1]. In such a table, the expected frequencies are typically those representing the null model: that the two proportions from the two rows (or columns) are equal. In a treatment/medical setting, we have the following 2 × 2 table.


Outcome


+−
Treatment+aca+c−bdb+d

a+bc+dN

There is a large body of literature on statistical modelling [2,3,4]. For our purposes, a model represents a means by which data may be produced or predicted, and specific hypotheses follow from the model. In this paper, we are concerned with the simplest of models when confronted by a 2 × 2 table representing binomial counts from two treatments. Most often, researchers are interested in whether the proportions/means of the two binomial counts from the two rows in the 2 × 2 table differ from the overall proportion/mean. More rarely, though with equal scientific validity, we see the issue of whether the variances of the two binomial counts differ from the average overall variance. As will be seen, the usual $X^{2}$ test statistic used to test proportions/means is inappropriate and should be restricted to test variances.

## 2. The Binomial Nature of the 2 × 2 Table

The outcome for each of the two treatments follows a discrete binomial distribution, where the number of ‘successes’ (positive outcomes) k in a sequence of n independent Bernoulli trials provides a probability p=k/n and a mass function for random variable X
$$P\left(X=k\right)=\frac{n!}{k!\left(n-k\right)!}p^{k}{\left(1-p\right)}^{n-k}$$
where k=0, 1, 2, ..., n. The likelihood of p for the sequence would be
$$L=C\times p^{k}{\left(1-p\right)}^{n-k}$$
where C is an arbitrary constant to scale the function so that it has a maximum of 1. In a likelihood ratio, the constant cancels out, as it is the same for both models.

We can envisage two models in this context, one unrestricted and the other restricted [5]. The unrestricted model is that k events out of a total number of n trials are binomial independent and identically distributed (i.i.d.) with a probability 0≤p≤1 taken from a distribution that is uniform and random. The mean and variance for the binomial distribution are np and npq, respectively, where q=1−p. For the first row, we can refer to use subscripts 1 and 2, respectively, for terms such as $n_{1}$, $n_{2}$, $p_{1}$, $p_{2}$, $q_{1}$, and $q_{2}$.

For the whole table, the proportion of successes would be those with a positive outcome
$$\pi =\frac{\left(a+b\right)}{N}$$

The mean being Nπ and variance Nπ(1−π).

Typically, the null model would specify that the two row proportions $\pi_{1}$ and $\pi_{2}$ are equal, and the same as the overall proportion
$$\pi =\pi_{1}=\frac{a}{c}=\pi_{2}=\frac{b}{d}$$

Consequently, the null model specifies that the two column proportions $\pi_{3}$ and $\pi_{4}$ are also equal
$$\pi_{3}=\frac{a}{b}=\pi_{4}=\frac{c}{d}$$

Alternatively and equivalently, we can talk about means, which was the preferred term used by Edwards in this context [6] (P182–191). For example, both the row mean number of positive outcomes would equal the overall mean number μ.

Unless specified *a priori*, the expected frequencies under the null hypothesis can be determined from the marginal totals. The degree of mismatch between the observed and the expected gives us the so-called test of association or independence. Tests can be made on the equality of the marginal totals (main effects), although these are usually of little or no interest.

The simplest situation of a 2 × 2 table is considered here, but the conclusions generalize to larger contingency tables, including multi-way tables.

## 3. Equations for $X^{2}$, Log Likelihood Ratio, and Likelihood Ratio Tests

The restricted model provides the null hypothesis specifying that the two binomial rows are from the same population. For the 2 × 2 table mentioned at the beginning, the expected value for the first cell ($a=O_{11}$) would be
$$E_{11}=\frac{\left(a+b)(a+c\right)}{N}$$

The Pearson $X^{2}$ statistic [7] can be calculated from a 2 × 2 table directly using
(1)$$X^{2}=\sum_{i=1}^{2}\sum_{j=1}^{2}\frac{{\left(O_{ij}-E_{ij}\right)}^{2}}{E_{ij}}$$
for two rows and two columns, where we denote the observed count in the ith row and jth column as $O_{ij}$ and the expected value in the ith row and jth column as $E_{ij}$. The expected values are either calculated from the marginal totals (for the null model) or are assigned according to a specified model (such as one suggested *a priori* by Mendelian inheritance). The distribution has $\left(i-1\right)\times \left(j-1\right)=1$ degree of freedom (*df*).

The log likelihood ratio is calculated for the two models. The first model is that the binomials in the two rows are independent and their means estimated from the data. The second model is that the two independent binomials have a common mean $\mu =\mu_{1}=\left(a+c\right)\pi_{1}=\mu_{2}=(b+d)\pi_{2}$. As earlier, the expected values are either calculated from the marginal totals or are assigned according to a specified model. Assuming the former model, we obtain the likelihood ratio for the first cell
$${LR}_{11}={\left(\frac{a\times N}{\left(a+b)(a+c\right)}\right)}^{a}$$
hence for the whole table, we obtain
$$LR=\frac{a^{a}b^{b}c^{c}d^{d}N^{N}}{{\left(a+b\right)}^{a+b}{\left(c+d\right)}^{c+d}{\left(a+c\right)}^{a+c}{\left(b+d\right)}^{b+d}}$$

A form of this equation was first derived by Sir Ronald Fisher [8] (P 128), and the full form given by Edwards [9]. The log likelihood ratio (support) for the first cell is
$$S_{11}=a\times \ln\left(\frac{a\times N}{\left(a+b)(a+c\right)}\right)=O_{11}\times \ln\left(\frac{O_{11}}{E_{11}}\right)$$

So that for the whole table, the log likelihood ratio sums the values for all cells
(2)$$S=\sum_{i=1}^{2}\sum_{j=1}^{2}\left\{O_{ij}\times \ln\left(\frac{O_{ij}}{E_{ij}}\right)\right\}$$

The analysis allows for cells to contain zeros, and calculations are performed using the convention that 0×ln⁡0=0. Here, *S* represents the term *support*, which was first defined by Harold Jeffreys as the natural logarithm of the likelihood ratio for one hypothesis versus another [10]. This scale represents the relative evidence and ranges from −∞ to +∞, with zero indicating no evidence either way. This is used in the likelihood approach [6,11,12], see later.

2S approximates to the χ^2^ distribution [13]
(3)$$2\sum_{i=1}^{2}\sum_{j=1}^{2}\left\{O_{ij}\times \ln\left(\frac{O_{ij}}{E_{ij}}\right)\right\}\;\dot{\sim \;}\chi^{2}$$

This is the form of the equations used for LRT analysis, where the right-hand tail area of the χ^2^ distribution is consulted for significance testing.

For the goodness of fit test where there is a single dimension of classification, Fisher [14] (P 357) (and noted by Woolf [15]) gave the expansion for support *S*
(4)$$\sum_{i=1}^{k}\left\{O_{i}\times \ln\left(\frac{O_{i}}{E_{i}}\right)\right\}=\sum_{i=1}^{k}\left[\frac{{{(O}_{i}-E_{i})}^{2}}{2E_{i}}-\frac{{{(O}_{i}-E_{i})}^{3}}{6E_{i}^{2}}+\frac{{{(O}_{i}-E_{i})}^{4}}{12E_{i}^{3}}-\cdots \right]$$
where *k* is the number of categories. The leading term in the expansion is of course half Pearson’s criterion. When the discrepancy between observed and expected relative to expected is small across all categories then
$$2\sum_{i=1}^{k}\left\{O_{i}\times \ln\left(\frac{O_{i}}{E_{i}}\right)\right\}≅\sum_{i=1}^{k}\frac{{{(O}_{i}-E_{i})}^{2}}{E_{i}}$$

Naturally, this extends to the addition of a second dimension for a contingency table. Again, we see the apparent relationship between the log likelihood ratio and the $X^{2}$ statistic. However, this relationship is deceptive as each is independently assessing a different parameter. One parameter is concerned with whether the ***proportions*** are equal to each other. The second is concerned with whether the ***variance*** of the two proportions differs from the overall average variance.

## 4. The LRT to Test for Proportions

Equation (3), given earlier, is known as the likelihood ratio test (LRT, also known as the G-test [15]). There are several advantages of using the LRT over the $\chi^{2}$ test, including better performance when expected numbers are small [16,17] and better theoretical grounding [14,18]. The LRT has the highest power among other competing tests according to the Neyman–Pearson lemma [19]. It avoids the $\chi^{2}$ test’s restriction that expected values should not be less than 5. The LRT is also computationally easier to calculate [20]. An important advantage of using the log likelihood ratio (Equations (2) and (3)) is that sub-tables within a large dimensional table can be independently analyzed, and that components from the sub-tables sum precisely, unlike in a χ^2^ analysis [21]. The same equations are employed for log-linear analyses of multi-way tables.

Use of the LRT to test proportions is non-controversial and is accepted as an appropriate way to test proportions. The LRT is based upon the likelihood of the data under a specific hypothesis (usually the null) relative to maximum likelihood estimates. Edwards (P 191) gives the derivation of Equation (2) for the log likelihood ratio from first principles [6]. As such, it represents a test of two binomial proportions. If the observed frequencies exactly match the expected frequencies, then *S* = 0, indicating no evidence of a difference from the null hypothesis.

The LRT assesses the equality of the model proportions, but it does not test the overall model variance.

## 5. The $\chi^{2}$ Test to Test Variance

There has been considerable confusion over the meaning of small $X^{2}$ statistics in categorical data analysis [22,23]. This occurs when there is little variation between observed and expected values. The $X^{2}$ statistic has been used to assess the variance of the model [16]. Specific use has been made in the analysis of Mendel’s results of plant hybridization experiments [24,25]. Actually, in his analysis, Edwards used X, rather than its square $X^{2}$, which had the advantage of preserving the direction of effect [25].

We can start from the first principles and consider the simpler binomial situation [25]. In testing the variance, the unrestricted model is the same as that used for testing the proportions/means, namely positive outcomes are binomial i.i.d. with a probability 0≤p≤1. For the restricted model, x is normally distributed and µ is fixed and part of the model. The alternative hypotheses concern different values for σ^2^, so that all the information we need is given by the $X^{2}$ statistic with 1 *df*
(5)$$X_{1}^{2}=\frac{{\left(x-\mu \right)}^{2}}{\sigma^{2}}$$

This is due to the following. For binomial events, μ=np, where n is the number of events in the sample and p is the probability of an event occurring. Hence $\sigma^{2}=npq$, where q=1−p, specifying a as the observed number of successes, and b=n−a will be the number of failures. Note: the a and b used here do not relate to the 2 × 2 table used earlier. Thus, a and b are the observed numbers in a binomial trial. Each approximates to a normal distribution, which improves as sample size increases. Our expected numbers are np and nq, respectively, since
$$X_{1}^{2}=\frac{{\left(a-np\right)}^{2}}{npq}\;=\frac{{\left(a-np\right)}^{2}}{np}+\frac{{\left(a-np\right)}^{2}}{nq}\;=\frac{{\left(a-np\right)}^{2}}{np}+\frac{{\left(b-nq\right)}^{2}}{nq}$$

Each of these last two terms representing the successes and failures, respectively, which is
$$=\sum \left\{\frac{{\left(observed-expected\right)}^{2}}{expected}\right\}$$

The expectation is precisely 1 because $E{\left(a-np\right)}^{2}=npq$. This is so because, *by definition*, npq is the variance. Knowing $\sigma^{2}=npq$ means that the expectation of the sum of the two terms for successes and failures will be 1. Moreover, if χ^2^ terms are independent of each other, then the expectation of their sum will exactly equal the *df*. This proof confirms the use of the $\chi^{2}$ test to test the variance in a contingency table. An alternative proof of this is given by Edwards [6] (P 182–183).

Hence, the $\chi^{2}$ test should be reserved for testing if the variance in the model is too low, as in determining whether the observed values match too closely the expected values. The data, then, appear to be ‘too good to be true’. This is not routinely of interest, but it can, for example, be important is assessing data integrity [25,26,27].

## 6. Evidential Support for the Variance

Using the likelihood approach, Edwards specifically derived an equation to test the variance in categorical data analyses [6]. From the previous section we know that the variance is assessed by the $X^{2}$ statistic. The distribution of $X^{2}$ on null hypothesis is
$$\frac{1}{2\sqrt{2\pi }X}e^{-\frac{X^{2}}{2}}d(X^{2})$$

Supposing the variance on the alternative hypothesis is $v^{2}\sigma^{2}$, then $X^{2}$ will be reduced by $v^{2}$ and distributed as
$$\frac{1}{2\sqrt{2\pi }vX}e^{-\frac{X^{2}}{{\;2v}^{2}}\;}d(X^{2})$$

For a given $X^{2}$, the likelihood ratio of the alternative hypothesis to the null is
$$LR=\frac{e^{-\frac{X^{2}}{{\;2v}^{2}}\;}}{ve^{-\frac{X^{2}}{2}}}$$

Taking logarithms to obtain the support
$$S=-\frac{1}{2}\ln v^{2}-\frac{X^{2}}{2v^{2}}+\frac{X^{2}}{2}$$

The best supported value of $v^{2}$ is $X^{2}$ as this value maximizes S. The maximum improvement in support is then
$$S=-\frac{1}{2}\ln X^{2}-\frac{1}{2}+\frac{X^{2}}{2}$$

This is the case for 1 *df*. As shown by Edwards [6] (P 186–187), this extends to multiple known means and hence greater *df*. This produces the general equation for support that can be used to test the variance ($S_{var}$) for any calculated $X^{2}$ statistic and its associated *df* [11]
(6)$$S_{var}=\frac{df}{2}\left(\ln\frac{df}{X^{2}}\right)-\frac{1}{2}(df-X^{2})$$

This can test whether the variance is larger or smaller than we would expect in the 2 × 2 table (here with 1 *df*). Typically, this would be used to test whether the data are too good to be true, i.e., the observed data fit expected too well (e.g., suggestive of data integrity issues, as in Mendel’s data). In the evidential approach, it can be used to assess the strength of evidence for lower-than-expected variance by referring to Table 1 given at the end. If all the observed values exactly match the expected values, then this equation would give $S_{var}=\infty$. Equation (6) may be more useful than Equation (1) since it calculates support for when $X^{2}$ test statistics approach 0 (i.e., when the variance is less than expected and the observed data are too good to be true). Multiplying $S_{var}$ by 2 then gives a $X^{2}$ statistic, and the *p* value accessed in the usual way by reference to the right-hand tail of the distribution. The $X^{2}$ test Equation (1) can also be used, but by referencing the left-hand tail of the distribution (note: if all observed = expected, then $X^{2}$ = 0 and left-hand tail *p* = 0).

## 7. Comparing the Three Equations

We can compare the performance of the three equations if we fix the all the expected frequencies to the same value assuming that all four marginal totals are 100 ($E_{ij}=50$, $\sum_{i=1}^{2}\sum_{j=1}^{2}E_{ij}=200$) and vary the observed values (which also maintain all the marginal totals to 100). The choice of 100 for the marginal totals is sufficiently large to simulate the situations typically encountered in practice. Each cell in the simulation varies from 1 to 99, so that a sufficient range of proportions is explored. The use of continuity corrections, though important [28], are not pertinent to the main issues explored by this paper, and can be examined in future work. Figure 1 shows the plot of LRT (2×S), $\chi^{2}$ test and $2\times S_{var}$ (to convert to approximate to the $\chi^{2}$ distribution). The observed value for the top left cell in the 2 × 2 table (cell *a*) varies from 1 to 99. When observed and expected values are close to each other, the LRT and $X^{2}$ tests produce almost identical results. Between observed values of 38 and 62, the $X^{2}$ statistic differs by less than 1% from the LRT statistic. In contrast, the $2\times S_{var}$ varies from the other two statistics, although it more consistently follows the $X^{2}$ statistic. As the discrepancy between the observed and the expected increases, say, beyond the range of 30–70, a clear divergence is apparent between the statistics for $X^{2}$ and $2\times S_{var}$ and the LRT statistic. The LRT is less conservative, giving chi-square values that are higher than the other two statistics. Consistent with $2\times S_{var}$ measuring variance, it is closest to the $X^{2}$ test line. In order to examine the $2\times S_{var}$ statistic at a higher resolution, as the observed and expected values become closer to each other, a plot on an expanded scale from 40 and 60 is given in Figure 2. This shows that as the observed value approaches the expected value (50), the $2\times S_{var}$ statistic increases dramatically, signaling that the variance of these probabilities is smaller than expected.

By specifying expected values for the top left cell in the 2 × 2 table (cell *a*), we can then plot the calculated chi-square for LRT and $X^{2}$ test for a range of observed values from 1 to 99. Figure 3 shows this for different expected cell a values of 20, 30, 50, and 90 (see insets showing *E*). As the expected value approaches their extremes (20 and 90 plots), the LRT statistics for even more extreme observed probabilities become more liberal (higher calculated statistics, giving smaller *p* values) compared to the $X^{2}$ test statistics. In contrast, the LRT statistics become more conservative for observed probabilities that head towards 0.5 and beyond.

Figure 1 and Figure 3 show that as large differences between observed and expected values occur, the lines for LRT and $X^{2}$ diverge. This is because they are measuring different things. The former is measuring the fit of the proportions, while the latter is measuring the variance in the model. Typically, we are interested in whether the proportions match the expected proportions. Generally, we are less frequently interested in whether the data are too close to expected frequencies [22]. Therefore, we should normally be using the LRT rather than the $\chi^{2}$ test.

Should we be interested in whether the observed frequencies too closely match frequencies specified by a particular hypothesis, then we should use the $\chi^{2}$ or $S_{var}$ test.

## 8. Comparison between LRT and $\chi^{2}$ Test for Type I and Type II Errors

The results of the simulations shown in Figure 1, Figure 2 and Figure 3 were obtained assuming that all marginal totals were fixed and equal. Typically, in randomized controlled trials, only the rows in the 2 × 2 table are fixed. Even with two rows fixed, as is typical in randomized clinical trials, a Monte Carlo simulation results in a confusion of lines. This is because the horizontal axis attached to one cell of the one row becomes meaningless because the calculated statistics also depend on the cells in the other row. In order to systematically compare the performance of the $X^{2}$ and LRT statistics in terms of type I and type II errors, binomial probabilities were applied to the first cells in the two rows. Each row had a fixed number of 100, while the column rows were determined by the binomial probabilities. Binomial randomness was applied to the selected probabilities, with 50,000 Monte Carlo repetitions for each point plotted in Figure 4. The probabilities ranged from 0.005 to 0.095. To examine type I error rates, identical probabilities were selected for the first cell of both rows, while for type II error rates, the first cell was fixed at 0.1. A significance level of 5% was used. The $X^{2}$ and LRT analyses produced similar type I error rates (around 5%) until a binomial probability of 0.08. Figure 4 shows that the type I error rate increased for LRT compared to $X^{2}$ analyses, as the probability decreased below 0.08, peaking at probability 0.015. In a complementary fashion, the type II error rate decreased for LRT compared to $X^{2}$ analyses. The actual type II error rates were similar (around 94.4%) at a probability of 0.1. At binomial probability 0.05, the type II error rates were 71.8% and 72.9% for LRT and $X^{2}$ analyses, respectively. These figures changed to 34.5% and 38.7% for a probability of 0.025, and finally to 4.2% and 7.2% for probability of 0.005. A mirror image of these results were obtained using binomial probabilities at the upper end from 0.9 upwards.

## 9. Effects of Conditioning on the Marginal Totals

The current analyses shown in Figure 1, Figure 2 and Figure 3 used all marginal totals fixed. There is a rich history of controversy over the effects of conditioning on the marginal totals, discussed by [21] (P 94–96 and note 3.9). Although simulations not conditional on the marginal totals produce a greater range of *p* values, it is suggested that conditioning on fixed marginal totals eliminates nuisance parameters. Furthermore, the margin totals anyway contain little information relevant to the association of interest. From the analysis of the type I and II error rates shown in Figure 4, it is clear that while LRT analyses have increased type I error rates, they are less likely to make type II errors. The discrepancies between the LRT and $X^{2}$ analyses most obviously occur when expected values are near the extremes (binomial probabilities outwith the range 0.1 to 0.9). In these circumstances, the LRT is generally more powerful than $X^{2}$ analyses. However, this also means that when the null hypothesis is true, LRT analyses will obtain higher values for the statistics, and hence an increase in type I errors.

There is general agreement that exact tests are preferable for small samples [21]. Convincing and cogent arguments have been made for the use of unconditioned exact tests for 2 × 2 tables, where smaller *p* values are obtained, and are hence more powerful than conditional tests [30,31]. These are almost certainly the best tests to use for the 2 × 2 situation, although, even with the current computational power, difficulties arise with larger tables. The primary purpose of this paper was to compare two commonly used tests, LRT and $\chi^{2}$, and to explain that the $\chi^{2}$ analysis is misused in most circumstances, since it is concerned with variance rather than independence of variables.

## 10. Empirical Data

Let us consider a particular set of data. Imagine that a large study was conducted on 20 thousand patients; half received a treatment and half received placebo. From this study, we obtained this contingency table:

***Death******Survival***
*Treatment*80992010,000*Placebo*120988010,000
20019,80020,000

Despite the rather modest odds ratio of 0.66 indicating a protective effect of the treatment, the result was highly statistically significant because of the large sample size with $X^{2}$(1) = 8.08, *p* = 0.004 (using LRT, we obtain a similar result $X^{2}$(1) = 8.13, *p* = 0.004).

Now, a small modification was made to the treatment, and some thought it might not make a difference. A smaller replication study was conducted with a tenth of the number patients, giving the following data:

***Death******Survival***
*Modified treatment*39971000*Placebo*109901000
1319872000

The general effect was similar, but achieved a better odds ratio of 0.30. The usual analysis of these data would give a non-statistically significant result with the $\chi^{2}$ test: 

$X^{2}$(1) = 3.79, *p* = 0.051

But it would be statistically significant with the LRT:

$X^{2}$(1) = 4.00, *p* = 0.045

*Since we are interested in proportions here, we should use the LRT result*. 

Although we could claim that there was a statistically significant beneficial effect of the modified treatment in the smaller sample, we do not know if the result differs from the previous large-scale study that used the original form of the treatment. To do this, we need to first obtain the probabilities for each outcome based upon the original study

***Death******Survival****Treatment*0.0040.496*Placebo*0.0060.494

The expected frequencies generated from these probabilities using the smaller sample, *N* = 2000, can then be used to calculate our two test statistics.

The $\chi^{2}$ test is non-statistically significant:

$X^{2}$(1) = 3.49, *p* = 0.062

Again, the LRT is statistically significant, this time by a larger margin:

$X^{2}$(1) = 4.50, *p* = 0.034

Again, since we are interested in proportions (not variances), then we should use the statistically significant LRT result, which suggests that the modified treatment is even better than the original treatment. This result is predicted from Figure 3 (top left panel), where there was a low expected value of 8 and an even lower observed value of 3 (resulting in more statistical power for the LRT versus the χ^2^ test).

## 11. Using the LRT to Test Proportions/Means (the Typical Scenario)

Hence, both theoretical and empirical considerations indicate that the correct test to test independence in a 2 × 2 contingency table is the LRT. In this test, we are interested in the two binomial proportions (typically the null hypothesis is that they are equal).

Although Sokal and Rohlf may have been unaware of the distinction between testing for proportions or variance in categorical data, in their seminal textbook [32], they wrote about the advantages of the LRT (which they call the G test) in their first edition (P 550):
*“…as is explained at various places throughout the text, G has general theoretical advantages over X^2^, as well as being computationally simpler for tests of independence. It may be confusing to the reader to have two alternative tests presented for most types of problems and our inclination would be to drop the chi-square tests entirely and teach G only. …to the newcomer to statistics, however, we would recommend that he familiarize himself principally with the G-tests.”*

After this proselytizing, their third edition of 1995 [33] (P 686) merely states:
*“…as we will explain, G has theoretical advantages over* $X^{2}$ *in addition to being computationally simpler, not only by computer but also on most calculators.”*

The calculational advantages are minimal now that most statistical packages routinely calculate both statistics. It is all too easily argued that the differences seen here between the two tests are marginal and might make little difference. The counterargument is to say that researchers feel more comfortable using a theoretically justified technique that is statistically testing what they are actually out to test. Edwards (P 193) comments scathingly: “The original test [$\chi^{2}$] gave the ‘right’ answer, but for the wrong reason. This is, of course, an expected characteristic of a procedure which has stood the test of time; it is only when we examine the problem closely that we realise the difficulties.” [6].

A contender for the shortest ever paper in statistics [34] provides a killer argument against the use of $X^{2}$ versus LRT. Briefly, reproducing from that paper, we can rewrite Equation (3) from above as
$$=2\sum \left[E+(O-E)\right]\ln\left[1+\frac{O-E}{E}\right]$$

When the logarithm part is expanded, we obtain our $X^{2}$ statistic plus other terms:$$=\sum \frac{{\left(O-E\right)}^{2}}{E}+terms\;of\;the\;order\;\left[\frac{{\left(O-E\right)}^{3}}{E^{2}}\right]$$

However, the logarithmic expansion, known as the Mercator series, is only valid to calculate the $X^{2}$ statistic if $\left|O-E\right|<E$. If this is untrue, even for a single cell, then the $X^{2}$ statistic should not be computed. Violation of the inequality will happen if the specified expected value for one of the cells is less than half of the observed value or, more trivially, if one of the cells in the table contains zero counts. On the strength of this, a prominent textbook on the analysis of contingency tables determined to exclusively use LRTs rather than $\chi^{2}$ tests [35] (see P 72).

We therefore have the encouragements from Sokal and Rohlf (1969), the unimpeachable arguments from Edwards (1972), a killer argument from Williams (1976), and a prominent textbook by Everitt (1992) to use the LRT in our typical quest to test proportions. All unheeded, since the consensus position remains that the LRT and $X^{2}$ statistics are, in most circumstances, equivalent. Furthermore, there appears to be no awareness among either researchers or statisticians of the distinction between the analysis of proportions and of variance in a contingency table. The $\chi^{2}$ test continues to be used inappropriately.

The general belief that $\chi^{2}$ and LRT tests give much the same result is erroneous: they often differ. For example, using models with latent variables, one study concludes that the asymptotic distribution of the LRT statistic does not follow $\chi^{2}$ distribution [36]. Also, consider the analysis above (**10. Empirical data**) where results from LRTs and $\chi^{2}$ tests straddled the statistical significance line. This can matter, since marginally significant or non-significant effects will be incorrectly reported one way or the other in a proportion of these tests. Decisions over whether effects are present often depend on the 5% significance level, and as such will influence whether a paper is published, a medical treatment adopted, or a research grant awarded. This reinforces the argument that it is essential to use the correct test for the parameter of interest (be it proportions or variance).

## 12. How to Decide Whether We Are Interested in the Proportions/Means or the Variance

How do we know which parameter we are interested in? Edwards suggested a simple question the investigator should ask before doing the $\chi^{2}$ test: “if I get a small $X^{2}$ will I be suspicious about my null hypothesis?” [6] P 190.

*If the answer is ‘No’, then we are interested in the proportions.* This would mean that if the observed values closely match the expected, we are happy to report the finding as consistent with the specified (or null) hypothesis.

If ‘Yes’, then we are interested in the variance. This would mean that a close match between the observed and expected values arouse our suspicions that there must be something ‘wrong’ with the data. This may be due to a number of reasons: errors in the data, bias, data selection, cooking [37], or even outright fraud [38].

To reiterate, the choice of statistical test should depend upon which model parameter is being tested. If a test of proportion equality is required (the typical situation), then the LRT is appropriate. The LRT is calculated directly from the log likelihood ratio. The latter can be used directly for statistical inference; see below. If a test of variance is required (e.g., the data are too good to be true), then the $\chi^{2}$ test should be used. Both tests exploit the $\chi^{2}$ distribution to calculate *p* values for statistical significance.

A natural alternative, for both analyses, is to use the evidential approach, avoiding the *p* values and the pitfalls of significance testing.

## 13. The Evidential Approach Using the Log Likelihood Ratio S

A more consistent and coherent approach is to calculate the log likelihood ratio (S or $S_{var}$) from the categorical data according to the expected values suggested by the model. The evidential approach was endorsed by Fisher later in his life, when he wrote [8]
*“…it is important that the likelihood always exists, and is directly calculable. It is usually convenient to tabulate its logarithm…”*

This is “directly calculable” because there is no need to calculate the tail probability integral.

The evidential approach was strongly promoted by Edwards in 1972 [9] and an expanded edition published in 1992 [6]. Additional arguments and perspectives were supplied by Royall [39]. More recently, the approach has been strengthened by further elaboration in books, chapters, and articles [11,40,41,42,43,44,45,46]. The approach allows the investigator to provide support for one hypothesis over another. In contrast, significance testing focusses on one hypothesis to be rejected. Such tests have a constant probability of a type I error if the hypothesis is true, irrespective of sample size. This contrasts with the evidential approach where the relative support of hypotheses can be determined, and the probabilities of weak and misleading evidence for one hypothesis versus another, decreases as sample size increases [42,47]. Although the log likelihood ratio as an evidence function is optimal for the simple case presented by the 2 × 2 table here, more complicated situations (involving outliers and nuisance parameters for example) can be dealt with using more sophisticated evidence functions [46].

In the context of the present paper, the evidential approach would calculate *S* values. To test proportions (the typical situation), Equation (2) is used
$$S=\sum_{i=1}^{2}\sum_{j=1}^{2}\left\{O_{ij}\times \ln\left(\frac{O_{ij}}{E_{ij}}\right)\right\}$$

To test for the variance, Equation (6) is used
$$S_{var}=\frac{df}{2}\left(\ln\frac{df}{X^{2}}\right)-\frac{1}{2}(df-X^{2})$$

In most research, only the first of these would be required. The second equation would be used to assess whether the fit of the data to the model is closer than expected (implemented in statistical platform *jamovi* (version 2.4) (retrieved from https://www.jamovi.org accessed on 19 March 2024) in the module named *jeva* (https://blog.jamovi.org/2023/02/22/jeva.html accessed on 19 March 2024)). In this approach, the support values obtained would be evaluated by reference to Table 1 below giving relative strengths of evidence [11,48].

**Table 1 entropy-26-00375-t001:** Using the log likelihood ratio for statistical inference. The left column gives the log likelihood ratio (*S*). The next column shows the likelihood ratio (LR). The final column gives the interpretations for comparing two hypotheses. Negative values of *S* would indicate the evidence for *H*_2_ exceeded that for *H*_1_.

*S*	LR	Evidence for *H*_1_ vs. *H*_2_
0	1	No evidence either way
1	2.7	Weak evidence
2	7.4	Moderate evidence
3	20.1	Strong evidence
4	54.6	Extremely strong evidence
7	1096.6	More than a thousand to one
14	1.2 × 10^6^	More than a million to one

## 14. Concluding Remarks

For the same contingency table data, two different tests can be performed, depending on the specific intention of the researchers. Researchers must determine which parameter of the contingency table they are interested in testing: the proportions or the variance. Typically, researchers are interested in whether the binomial proportions differ, in which case, the LRT should be used. More rarely, researchers are interested in testing the variance, in which case the $\chi^{2}$ test should be used. The current use of the $\chi^{2}$ test to test proportions/means is flawed, representing widespread misuse of the test. A consistent and coherent approach to statistical inference is offered by support tests using the log likelihood ratio. Again, for contingency tables, the researcher must determine which parameter is of interest. Two different equations to calculate the log likelihood ratio (S and $S_{var}$) are available, one to test for proportions and the other to test for variance.

## Figures and Tables

**Figure 1 entropy-26-00375-f001:**
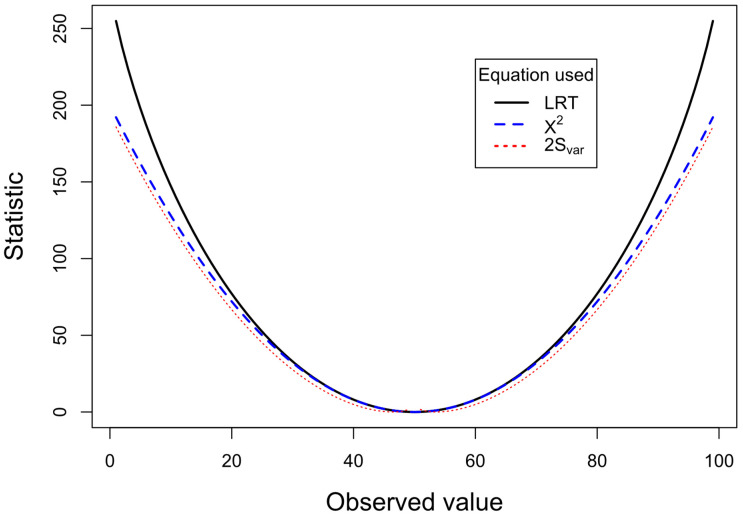
Calculated statistics plotted for 2 × 2 contingency table with *N* = 200 and all marginal totals fixed at 100. The observed values for one of the cells vary from 1 to 99, while expected values are 50 for all cells. The plot for the LRT statistic (Equation (3)) is shown by the black continuous line, $X^{2}$ test (Equation (1)) is shown by the blue dashed line, and the $2\times S_{var}$ (Equation (6)) is shown by the red dotted line. Applying Williams’s correction [29] to the LRT has a negligible effect and produces a line that closely overlaps with the LRT line.

**Figure 2 entropy-26-00375-f002:**
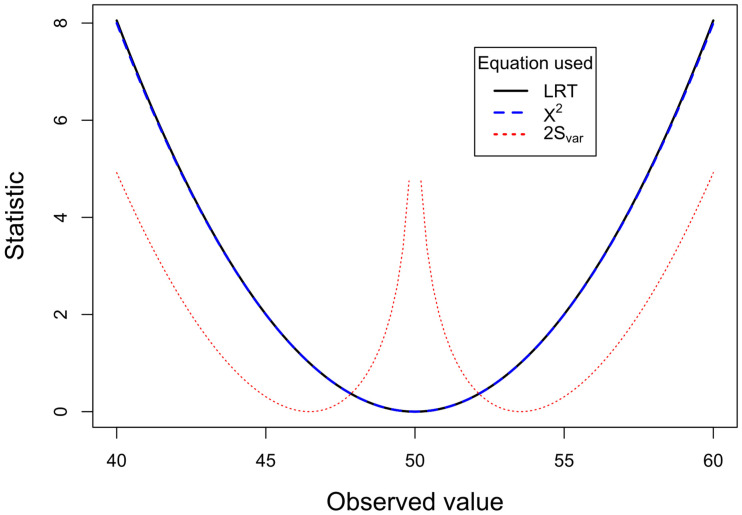
The same plots as in Figure 1 but on an expanded horizontal scale from 40 to 60. The LRT and $X^{2}$ test curves overlap very closely, reaching 0 when observed and expected values equalize at 50. In contrast, the $2\times S_{var}$ line increases dramatically as observed values approach expected (and is ∞ when they are equal).

**Figure 3 entropy-26-00375-f003:**
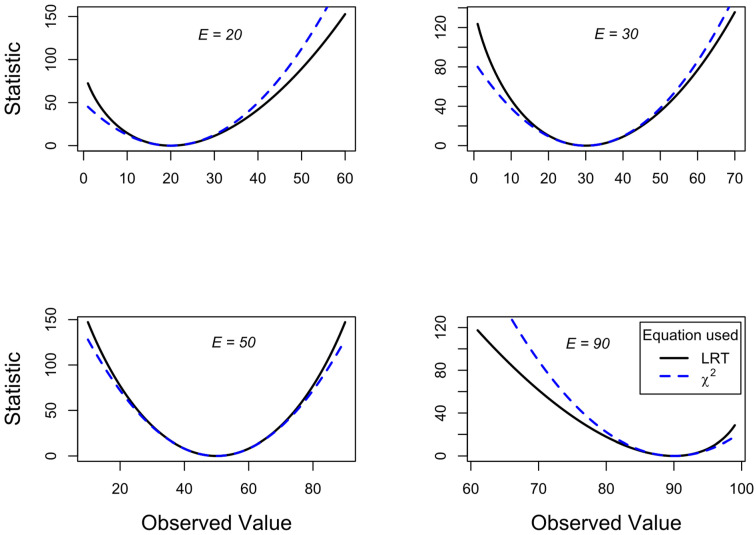
The statistic using LRT (Equation (3), black continuous line) and $X^{2}$ test (Equation (1), blue dashed line) are plotted for different fixed expected values (see inset expected *E* values from 20 to 90), where the observed values vary in each plot. Plots use 2 × 2 contingency table data where *N* = 200 and all marginal totals fixed at 100. The plot for *E* = 50 is similar to that shown in Figure 1, but without the $2\times S_{var}$ line. For display purposes, different scales are used for vertical and horizontal axes in each plot.

**Figure 4 entropy-26-00375-f004:**
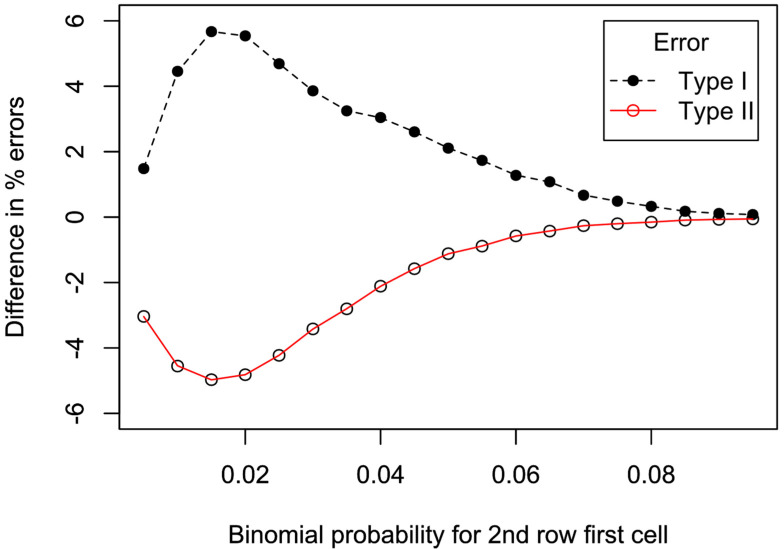
Difference in percentage of type I and type II errors for LRT—$X^{2}$ analyses. For type I errors (dashed line, filled circles), the same binomial probabilities were assigned to the first cell of each row and systematically changed from 0.005 to 0.095. For type II errors (continuous line, unfilled circles), the first cell was fixed with a probability of 0.1, while the binomial probability for the first cell of the second row was systematically changed from 0.005 to 0.095. Data were obtained by Monte Carlo simulations of randomly generated binomial numbers, with 50,000 repetitions for each point. R code for this and other figures is available in Appendix A.

## Data Availability

R code to produce all the figures is available in Appendix A.

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
