# Peer review of "Likelihood Ratio Test and the Evidential Approach for 2 × 2 Tables"

_entropy, 2024, doi:10.3390/e26050375_

Round 1

Reviewer 1 Report (Previous Reviewer 1)

Comments and Suggestions for Authors

This is a good paper that offers some adequate testing guidance to a general audience. 

Author Response

Thank you for your positive comments. I have improved the manuscript by rearranging the text so that it flows better and introduced a number of new headings to clearly distinguish different topics covered. At the end I have added Concluding remarks: 

For the same contingency table data, two different tests can be performed, depending on the specific intention of the researchers. Researchers must determine which parameter of the contingency table they are interested in testing: the proportions or the variance. Typically, researchers are interested in whether the binomial proportions differ, in which case the LRT should be used. More rarely, researchers are interested in testing the variance, in which case the χ2 test should be used. The current use of the χ2 test to test proportions is flawed, representing widespread misuse of the test. A consistent and coherent approach to statistical inference is offered by support tests using the log likelihood ratio. Again, for contingency tables, the researcher must determine which parameter is of interest. Two different equations to calculate the log likelihood ratio (S and Svar) are available, one to test for proportions and the other to test for variance.

Reviewer 2 Report (New Reviewer)

Comments and Suggestions for Authors

This paper considers analysis of 2x2 tables. The main concern is which test to use for testing the independence hypothesis. The literature on this topic is reviewed. Some simulation evidence is given. This leads the author to prefer the likelihood ratio test statistic over other statistics.

I find that the question of interest is not very well defined: What is the model?  What is the criteria for preferring a test? Is this uniform over parameters, sample size, data figurations? The simulation study has only one design. In the end I find that there is not sufficient evidence for me to think
differently about this problem than before.

Comments on the Quality of English Language

First 6 lines of intro not helpful for me. Is this a quote? If so a reference should be given.

Some sentences stop abruptly, e.g. line 37.

Author Response

Thank you for your constructive and insightful comments.

I have improved the manuscript by rearranging text, including new text, and inserting new headings to clarify the flow and make the focus of the article much clearer. I have highlighted new text. I have modified extensively the Abstract and have added a Concluding remarks section at the end.

I have added a section labelled How to decide whether we are interested in the proportions or the variance. This answers your question about the model of interest and the criteria for preferring one test over the other. 

The first few lines are a quote from a TV Series, reference to which I have now added at the end. The quote is concerned with the foundations for an argument. The relevance to my article is two-fold. First, my article is concerned with the flawed theoretical foundations of using the χ2 test for testing proportions (the usual scenario). The appropriate test is the likelihood ratio test. A second reference to theoretical foundations concerns the use of evidential statistics in contrast to statistical testing (with the issues of using p values). 

I have corrected the error at line 37, thank you.

Reviewer 3 Report (New Reviewer)

Comments and Suggestions for Authors

The author reviews the application of the chi-square test and the LRT in a 2*2 table.

I only have one question. Why don't you use a continuity correction? Not using a continuity correction in this analysis is a serious error, which invalidates the entire study.

The author should carry out a more appropriate bibliographic review on the topic. For example:

Martín-Andrés et al. (2005). On the validity condition of the chi-squared test in 2×2 tables. Test, 14 (1):99-128.

Author Response

Thank you for your comments.

I have improved the manuscript by rearranging text, including new text, and inserting new headings to clarify the flow and make the focus of the article very clear. I have highlighted new text. I have modified extensively the Abstract and have added a Concluding remarks section at the end.

Thank you for your comment on continuity corrections and reference which I have now included in the manuscript. In the legend Figure 1, I noted that using the correction suggested by Williams (1976) had negligible effect on the plotted line. Also, in the first paragraph under the heading Comparing the 3 equations, I have added: "The use of continuity corrections, though important (Andrés et al. 2005), are not pertinent to the main issues explored by this paper, and can be examined in future work."

Round 2

Reviewer 2 Report (New Reviewer)

Comments and Suggestions for Authors

It certainly helps that headings have been added throughout.

I continue to find that the question of interest is not very well defined. When using an LR test one needs to know what the unrestricted probability model and the hypothesis are. The abstract says that "the approach facilitates
the calculation of appropriate log likelihood ratios ... to test the proportions or to test the variance".  For the test for independence, the hypothesis is discussed on page 3, although the reader is left to guess what probability model the author has in in mind. The unrestricted model is not described.  Presumably, the author means i.i.d. binomial with freely varying probabilities? For the test of variance, the unrestricted probability model and the hypothesis are never defined.  

Usually it is a good idea to use LR tests. But it is rare that LR dominates other tests in finite samples uniformly over the parameter space.  Thus, one needs to formulate a criteria for preferring a test. And then investigate if a particular test is better uniformly over parameters, sample size and/or data figurations. With only one simulation study the interesting questions are not really addressed.  For instance, the sample size in the simulation is so big that we do not even see what William's finite sample correction is meant to do (Figure 1).

As I wrote before, I do not find that there is sufficient evidence for me to think differently about this problem than before.

Author Response

I address the comments in the order they were made. New changes are indicated by yellow highlighting in the uploaded MS.

  1.  I have provided descriptions of unrestricted and restricted models for the analyses of means and variances. For each of the analyses I have provided derivations of the relevant equations used. I have provided greater detail than that offered by Edwards (1992), making them more accessible to the reader.
  2.  Because it has not been appreciated that different analyses are appropriate to answer different questions, I have included a section on how the researcher should decide which analysis to use (How to decide whether we are interested in the proportions/means or the variance)
  3. The reviewer says: "then investigate if a particular test is better uniformly over parameters, sample size and/or data figurations." This is a fair point, the simulation data provided by the MS is very limited. That is because the main purpose was to address the issue primarily from a theoretical perspective, rather than to investigate relative performance over parameters.
  4.  As Edwards recognised over 50 years ago, the chi-square test should only be a test for variance, not proportions/means (for which the LRT is appropriate). The article is concerned with foundational issues. We should use the appropriate tool for a job, not a lesser one, even when that one appears to “have stood the test of time”. There should be no excuse based on pedagogy, history or culture – we should simply use the correct procedure. This applies to the correct analysis of contingency tables depending on our question. The piece is also concerned with promoting the evidential approach. I have attempted to do this by presenting a number of arguments. The most important point of the paper is the misuse of the chi-square statistic to analyse categorical data. Under the section The chi-square test to test variance I have provided a simple proof that the chi-square statistic is concerned with assessing the variance. I follow this up with an expanded derivation of the equation required to provide the support (Svar) for the variance. This formula incorporates the chi-square statistic (equation 6). Since the chi-square statistic is only appropriate to assess the variances, the LRT should be used to assess the means. There is also Williams's ‘killer’ argument (Williams 1976b) against the use of chi-square which shows that the logarithmic expansion is only valid if |O - E| < E . I compare the performance of the 3 equations (LRT, 2Svar, chi-square) in a limited simulation. This is followed by a comparison of type I and type II errors (since this was requested by a reviewer in round 1). I end the piece arguing that the optimal strategy would be to use the evidential approach, and provide the two equations (those previously derived) in order to perform the relevant analysis (for means, or for variances depending on the question).

Reviewer 3 Report (New Reviewer)

Comments and Suggestions for Authors

It is incomprehensible that the author does not consider studying continuity corrections. I have nothing more to say.

Author Response

Dear reviewer, I inserted a reference that you had recommended (Andrés et al. 2005), for which I thank you.

The issue of continuity correction (as its name suggests) is concerned with providing a more accurate tail probability integral for a discrete distribution by approximating it to a continuous distribution. This is most important for small samples. Since this is very tangential to the main points of the paper, I have not addressed it further.

This manuscript is a resubmission of an earlier submission. The following is a list of the peer review reports and author responses from that submission.

Round 1

Reviewer 1 Report

Comments and Suggestions for Authors

Reviews: Entropy-2759083

Which Statistical Test should we use for 2 x 2 Tables?

This paper reinforces a statistical argument for using the LR test to test for the association in a 2 x 2 table. And provides convincing level of detail of the supporting arguments. Of particular interest is the authors elaboration of the historical development on the topic. In addition to the interesting historical overview, this paper should appeal to practitioners in disciplines where the distinction between the Chi-square tests (CST) and the likelihood ratio test (LRT), less well understood, as well when the preference for LRT over CST is not common knowledge.

In many fields, statistical models are often applied in situations like this (i.e., loglinear or logit models).  Specifically, for the 2 x 2 table, the null hypothesis to be tested concerns independence of the two variables comprising the table. In this case, quantity (following Edwards) is called S. In statistical modeling software this is usually called the deviance, as it reflects a departure from the data, or the saturated model (both CST and LRT are reported in some software). However, modeling generally focuses on assessing the strength of the association (i.e., the interaction) via tests on the log odds ratio that has good asymptotic properties. The relevant parameter in the 2 x 2 table is the cross-product ratio (or specifically the log odds ratio). So why not a simple z-test on the single parameter. Post estimation, such models can also produce marginal predictions of the probabilities for the focal variable for each category of the other, thus allowing tests of difference in proportion or any linear or nonlinear transformation of the raw coefficients for that matter.

Critique:

The quantity S is referred to as a likelihood ratio test (LRT). It is a specific one, that is equivalent to reported residual deviance for models fit to aggregated data tables. For the 2 x 2 table the deviance (and S) tests the adequacy of the current model against the saturated model. Other LRT compare the current model to the null model, or in higher dimensions, the current model against nested models. Confusion often arises in models fit to different data structures such as individual level binary data, where the log L of the saturated model is 0, making the deviance simply -2logL, which has questionable utility for evaluating fit. A researcher analyzing the 2x2 contingency table gets the same association estimate as the researcher analyzing individual binary data, but the reported deviances have different meaning in these two contexts.

Long story short, I think the paper would benefit from equating S with the more familiar deviance (unless I am missing something). To be fair, I believe the paper references G or perhaps G-squared, which are other names for the deviance.

Author Response

Thank you for the positive review and constructive comments.

  1. The reviewer states that the quantity S (following Edwards and used in the paper) is the same as the deviance. This is not correct. S is the log likelihood ratio while the deviance is -2 × log likelihood ratio. This is only true for non-Gaussian models such as Poisson and binomial, see Aitkin, M. (2022). Introduction to Statistical Modelling and Inference, CRC Press. P 188 section 14.4.
  2. The reviewer suggests that for the 2 × 2 table the log odds ratio assessed with a simple z test on a single parameter would be preferable. This is an elegant suggestion, and the author agrees that in many scenarios this is a useful statistic. However, this statistic does not generalize to larger contingency tables, while S and the LRT do. Also, when one cell in the table is 0, or there are 0s in different columns, then the Haldane-Anscombe correction must be applied. When both cells in one column of the table are 0, the odds ratio and log odds ratio (and the chi-square) cannot be calculated. In all these situations the LRT can be calculated. In the most extreme situation where both cells in one column are 0, both the LRT and S can still be calculated, and both are 0. This is achieved using the convention that 0 × ln(0) = 0. A section paraphrasing the above has been added to the Discussion section.
  3. The final point suggests that the paper would benefit by equating S with deviance. As explained above, these quantities are different.

Reviewer 2 Report

Comments and Suggestions for Authors

The author compared two commonly used tests for 2x2 tables in the setting of randomized clinical trials. However, both these tests assume all four margins are fixed, which is not the case for randomized trials. The simulation is done in a very restricted setting (all margins are equal and fixed) and the comparison is on the test statistics based on observed cell counts. In reality for randomized trials, only row totals (sample size of two groups) will be fixed.   A more appropriate simulation should be done under null and alternative hypotheses assuming group sizes fixed only and compare the type I error and power of the two tests under different settings. Note both tests are asymptotic tests that are not appropriate for sparse tables. Besides well known conservative Fisher's exact test that also assumes all margins are fixed, multiple unconditional exact tests (assume only row or column totals fixed) have been proposed. From theoretic aspect, unconditional exact test are better and available from commonly used software (such as Barnard test available in SAS and multiple unconditional exact tests available in R exact.test (Exast)). 

Author Response

Thank you for your positive review, comments and helpful suggestions.

  1. The reviewer states that the simulations were done in the restricted setting of all marginal totals being fixed. As noted by the reviewer, Fisher’s exact test assumes fixed marginal totals to calculate the p value. The discussion by Agresti (2013, p94-96 and note 3.9) indicates that there is controversy over conditioning by many authors: controversy over conditioning includes Barnard (1945, 1947, 1949, 1979), Berkson (1978), Cheng et al. (2008), Fisher (1956) etc. It is suggested that conditioning on fixed marginal totals provides a simple way to eliminate nuisance parameters, and that they anyway contain little information relevant to the association of interest. Simulations not conditional on the marginal totals produce a greater range of p values. Fixed marginal totals were used in the current paper to allow plots to be made of the ordered changes in statistics as the observed value in only one cell is systematically changed from low to high values. Even with two rows fixed, as is typical in randomized clinical trials, a Monte Carlo simulation results in a confusion of lines, since the horizontal axis attached to one cell of the first row becomes meaningless because the calculated statistics also depend on the cells in the second row.
  2. The reviewer helpfully suggests simulating with rows fixed and comparing type I and type II error rates between X2 and LRT. In these simulations the effects of the binomial probabilities assigned to the first cell in the two rows were observed. If these probabilities were the same, then the type I errors could be examined. Monte Carlo simulations found that X2 and LRT produced virtually identical percentages of errors (close to 5% using significance level of 5%) across probabilities from 0.1 to 0.5. More important, as indicated in Figure 3, the percentage of type II errors were greater for X2 than LRT when low (or high) expected values were examined (binomial probabilities outwith 0.1 and 0.9). The new Figure 4 shows the results of Monte Carlo simulations in the Results section, and discussed in the 2nd paragraph of the Discussion section.

Reviewer 3 Report

Comments and Suggestions for Authors

I have read the first one and a half page of the manuscript. Unfortunately, there are many weaknesses and errors, see my comments below.

1.

To my knowledge, the most up to date and comprehensive comparisons of methods for testing in 2x2 tables are found in the following: (Fagerland, Lydersen, & Laake, 2017; Lydersen, Fagerland, & Laake, 2009). You ought to study these publications and refer to at least one of them if you write a manuscript on this topic.

2.

The first sentence in the abstract could rather be

Analyses of 2x2 crosstables are common, not least because they can provide risk difference, risk ratio, or odds ratio in medical research.

3.

Second sentence in abstract:

There are many versions of the X2 test.

4.

You write “… data appear to match too closely a particular hypothesis ...“. This is unclear and possibly not correct.

After “better call Saul”:

5.

Delete the first sentence.

6.

The second sentence is incorrect.

You could formulate it for example as

“The Pearson chi squared statistic for a 2x2 table is defined as (Pearson 1900):

7.

First sentence after equation 1:

i and j are running indexes. There are 2 rows and columns (or generally for example r rows and 2 columns).

8.

The distribution is only asymptotically chi square distributed with 1 degree of freedom.

9.

You write:

“There is general agreement that exact tests are preferable for small samples (Agresti 2013), for example Fisher’s exact test.”

There has been a vivid dispute about Fisher’s exact test. In small samples, an exact unconditional test or a mid-p test has better properties. See for example (Lydersen et al., 2009) or (Fagerland et al., 2017)

Fagerland, M., Lydersen, S., & Laake, P. (2017). Statistical Analysis of Contingency Tables.: Chapman and Hall/CRC.

Lydersen, S., Fagerland, M. W., & Laake, P. (2009). Recommended tests for association in 2 x 2 tables. Stat. Med, 28(7), 1159-1175.

Comments on the Quality of English Language

The English language is good, as far as I can judge from the part I have read.

Author Response

Thank you for the helpful and positive comments and suggestions.

  1. Pointing me to the book and article by Fagerland et al was most useful. I have added them in the references, and added the following in the 3rd paragraph of the Discussion:
    Convincing and cogent arguments have been made for the use of unconditioned exact tests for 2 × 2 tables, where smaller p values are obtained, and are hence more powerful than conditional tests (Lydersen, Fagerland, and Laake 2009; Fagerland, Lydersen, and Laake 2017). These are almost certainly the best tests to use for the 2 × 2 situation although, even with current computational power, difficulties arise with larger tables. The primary purpose of this paper was to compare two commonly used tests, LRT and χ2, and to explain that the χ2 analysis is misused in most circumstances, since it is concerned with variance rather than independence of variables.” See also point 9 below.
  2. Thank you, I have now added the other statistics you mention in the Abstract.
  3. I have modified the sentence to: “A χ2 analysis is most often used, although some researchers use a likelihood ratio test (LRT) analysis.”
  4. The sentence as it stands is correct, as it is claimed that χ2 is only appropriate where the data match too closely a model being tested (for example that suggested by Mendelian inheritance, or in other situations where the data look “too good” possibly due to data fraud. The issue is explained in more detail, specifically just after equation (6).
  5. Done, thank you.
  6. Done, thank you.
  7. Modified as suggested, thank you.
  8. I am unsure where in the manuscript this comment is directed. Please clarify.
  9. I have reduced the sentence to read: “There is general agreement that exact tests are preferable for small samples (Agresti 2013).”
    I have also added reference to this issue in point 1 above, where I have added to the Discussion section.